

# Quercetin prevents chronic kidney disease on mesangial cells model by regulating inflammation, oxidative stress, and TGF-β1/SMADs pathway

Wahyu Widowati[1], Sijani Prahastuti[1], Rita Tjokropranoto[1], Philips Onggowidjaja[1], Hanna Sari Widya Kusuma[2], Ervi Afifah[2], Seila Arumwardana[2], Muhamad Aldi Maulana[2] and Rizal Rizal[2,3]

[1] Faculty of Medicine, Maranatha Christian University, Bandung, West Java, Indonesia
[2] Biomolecular and Biomedical Research Center, Aretha Medika Utama, Bandung, West Java, Indonesia
[3] Biomedical Engineering, Department of Electrical Engineering, Faculty of Engineering, University of Indonesia, Depok, West Java, Indonesia

## ABSTRACT

**Background**. Chronic kidney disease (CKD) happens due to decreasing kidney function. Inflammation and oxidative stress have been shown to result in the progression of CKD. Quercetin is widely known to have various bioactivities including antioxidant, anticancer, and anti-inflammatory activities.

**Objective**. To evaluate the activity of quercetin to inhibit inflammation, stress oxidative, and fibrosis on CKD cells model (mouse mesangial cells induced by glucose).

**Methods and Material**. The SV40 MES 13 cells were plated in a 6-well plate with cell density at 5,000 cells/well. The medium had been substituted for 3 days with a glucose-induced medium with a concentration of 20 mM. Quercetin was added with 50, 10, and 5 µg/mL concentrations. The negative control was the untreated cell. The levels of TGF-β1, TNF-α, and MDA were determined using ELISA KIT. The gene expressions of the SMAD7, SMAD3, SMAD2, and SMAD4 were analyzed using qRT-PCR.

**Results**. Glucose can lead to an increase in inflammatory cytokines TNF-α, TGF-β1, MDA as well as the expressions of the SMAD2, SMAD3, SMAD4, and a decrease in SMAD7. Quercetin caused the reduction of TNF-α, TGF-β1, MDA as well as the expression of the SMAD2, SMAD3, SMAD4, and increased SMAD7.

**Conclusion**. Quercetin has anti-inflammation, antioxidant, antifibrosis activity in the CKD cells model. Thus, quercetin is a promising substance for CKD therapy and further research is needed to prove this in CKD animal model.

# INTRODUCTION

Chronic kidney disease (CKD) is a disorder in which the function of the kidneys is harmed. This is indicated by a lower glomerular filtration rate (GFR) or less than 60 mL/min per 1.73 m². CKD is a global health concern, with a 1.8% increase in frequency in Indonesia between 2013 and 2018 (*Webster et al., 2017*). The main causes of CKD are

Corresponding author
Wahyu Widowati,
wahyu_w60@yahoo.com

diabetes and hypertension. One of the characteristics of CKD is fibrosis. Renal fibrosis is a long-term, progressive condition that affects kidney function during CKD disease (*Chen et al., 2018a*; *Chen et al., 2018b*). The reason of renal fibrosis is that extracellular matrix (ECM) proteins such as collagens, fibronectin, and laminin accumulated in the diabetic nephropathy (DN) condition (*Zeng, Xiao & Sun, 2019*). Pathogenesis of fibrosis of CKD implicates oxidative stress and inflammatory reactions (*Popolo et al., 2013*). Inflammation and oxidative stress have been shown to result in the development of CKD (*Qian, 2017*). Nuclear factor-kappaB (NF-kB) and nuclear factor-erythroid-2-related factor 2 (Nrf2) were triggered by inflammation and oxidative stress (*Luo et al., 2021*).

Patients with CKD usually suffer from chronic inflammation (*Morena et al., 2002*), and have severely damaged antioxidant systems that gradually worsen with renal failure (*Libetta et al., 2011*). Tumor Necrosis Alpha (TNF-α) and Transforming Growth Factor β1 (TGF-β1) are important cytokines in the inflammation process. Meanwhile, malondialdehyde (MDA) is a valuable parameter for oxidative stress (*Webster et al., 2017*). SMADs are the primary signaling mechanism by which TGF-β1 mediates CKD (*Rapa et al., 2020*). Thus, in uremic syndrome, modulated inflammation cytokines and oxidative stress are of primary importance for treating CKD (*Rapa et al., 2020*).

The most common therapies to treat the progression of CKD are anti-inflammatory drugs, antioxidants, and renin-angiotensin system (RAS) inhibitors (*Zhou et al., 2015*). However, those therapies are not adequate to prevent the disease. As a result, scientists around the world are still looking for new and innovative therapies on CKD (*Li et al., 2019a*; *Li et al., 2019b*). Natural product is an alternative therapy for CKD and anti-fibrosis (*Chen et al., 2018a*; *Chen et al., 2018b*). A previous study showed that Epigallocatechin-3-gallate from green tea (EGCG) treatment could inhibit fibrosis by TGF-B/SMADs pathway, also known for having strong antioxidant and anti-inflammation properties (*Wang et al., 2015*). Emodin is a natural component of Rhei Radix et Rhizoma, a Chinese herbal medicine with anti-inflammatory, antioxidant, anti-fibrotic, and antibacterial properties (*Xu et al., 2021*). In Indonesian herbal medicine, several flavonoids have been effectively found from *Phyllanthus niruri*, such as quercetin, which has been shown to have significant antioxidant and chelating properties (*Rusmana et al., 2017*).

Quercetin is a major flavonoid content in most plant extracts (*Junior et al., 2018*). Quercetin is found in apples, red grapes, kales, broccoli, onions, berries, and cherries (*Xu et al., 2019*). Like other flavonoids, quercetin has various bioactivities such as antibacterial, antioxidant, anti-inflammatory, and antivirus activities (*Perdhana & Suzana, 2019*; *Prahastuti et al., 2019*). Quercetin has been found in previous research to protect the kidneys by reducing glucose levels (*Vera et al., 2018*). By enhancing the antioxidant capacity of cells, quercetin can help minimize oxidative stress (*Wu et al., 2017*). Moreover, by suppressing inflammatory factors, quercetin can lessen inflammatory stimulation (*Tan et al., 2020*). Quercetin inhibits the transformation of renal tubular epithelial cells, which can postpone fibrosis and modulate macrophage polarization (*Lu et al., 2018*; *Yang et al., 2018*). Moreover, it has been proven that quercetin could protect human renal proximal tubular cells against the toxicity of radiocontrast medium in a recent study (*Andreucci et al., 2018*). The previous study showed that quercetin could improve renal function and

reduce oxidative stress factors, serum level of Fibroblast Growth Factor 23 (FGF23), renal inflammation, and renal tubular damage on rat adenine-induction CKD model (*Yang et al., 2018*). However, further research is needed to see if quercetin can help in CKD treatment. Present study evaluated the potency of quercetin for CKD therapy with *in vitro* study in a cell model of CKD by modulating the production of TNF-α, TGF-β1, and MDA as inflammation cytokines, fibrosis, and oxidative stress marker and SMAD2, SMAD3, SMAD4 and SMAD7 as inflammation and fibrosis genes.

## MATERIALS AND METHODS

### Cell culture and CKD treatment

Cell lines of mouse mesangial kidney SV40 MES 13 cells (ATCC® CRL-1927TM) were received from Aretha Medika Utama, Biomolecular and Biomedical Research Centre, Bandung, West Java, Indonesia, and plated in a 6-well plate (3516; Costar, Wujiang, Jiangsu, China) with a total of 5,000 cells per well. The cell media used were Dulbecco's Modified Eagle Medium (DMEM) (L0103-500; Biowest, Riverside, MO, USA) and F12-K Mix Nutrient (L0135-500; Biowest, Riverside, MO, USA) with a ratio of 1:3 which had been substituted for 3 days with a glucose-induced medium with a concentration of 20 mM. Quercetin (Q4951-10G; Sigma Aldrich, Saint Louis, MO, USA) was added with 50 and 10, and 5 μg/mL in concentration. The positive control was mesangial cell induced by glucose without using quercetin, and negative control was the untreated cell (*Sandhiutami et al., 2017*; *Widowati et al., 2018*; *Hidayat et al., 2022*).

### Total protein assay

The bovine serum albumin (BSA) (A9576, Lot. SLB2412; Sigma, St. Louis, MO, USA) 2 mg was used to get standard solution. The 20 μL of standard solutions, quercetin, and 200 μL Quick Start Dye Reagent 1× (5000205; Bio-Rad, Hercules, CA, USA) were added briefly into each well. The plate was then incubated at room temperature for 5 min. The absorption was measured at 595 nm with the microplate reader (Multiskan GO Microplate Spectrophotometer; Thermo Fisher Scientific, Waltham, MA, USA) (*Lister et al., 2020*).

### TGF-β1, TNF-α, MDA using ELISA method

The treated mesangial cells condition medium was used as the sample. Mouse TGF-β1 ELISA Kit (E-EL-M0051; Elabscience, Houston, Texas, USA), Mouse TNF-α (E-EL-M0049; Elabscience, Houston, Texas, USA), and MDA (E-EL-0060; Elabscience, Houston, Texas, USA) according to the protocol of manufacture with modified method were conducted. The absorbance was read at 450 nm using microplate reader (*Sandhiutami et al., 2017*; *Widowati et al., 2018*; *Lister et al., 2020*; *Ginting et al., 2021*; *Prahastuti et al., 2019*; *Hidayat et al., 2022*).

### Gene expression of SMAD7, SMAD2, SMAD3, SMAD4 using RTPCR method

TRI reagent (R2050-1-200; Zymo, Irvine, CA, USA) was used to extract the RNA of cell ($1 \times 10^6$ cell/mL) and RNA isolation KIT (R2073, Zymo, Irvine, CA, USA) was used to purify the RNA. These procedures were in accordance with the manufacturer's protocol.

**Table 1  Primer sequences of GAPDH, SMAD2, SMAD3, SMAD4, SMAD7 genes.**

| Gene | Primer sequence (5′–3′) Upper strand: sense Lower strand: antisense | Product size (bp) | Annealing (°C) | Cycle | Reference |
|---|---|---|---|---|---|
| GAPDH | TCAAGATGGTGAAGCAG ATGTAGGCCATGAGGTCCAC | 217 | 59 | 40 | NCBI Reference Sequence: NM_001289726 |
| SMAD2 MOUSE | ATTACATCCCAGAAACACCAC TAGTATGCGATTGAACACCAG | 196 | 59 | 40 | NCBI Reference Sequence: NM_001252481.1 |
| SMAD3 MOUSE | GTAGAGACGCCAGTTCTACCT CATCTTCACTCAGGTAGCCAG | 178 | 59 | 40 | NCBI Reference Sequence: NM_016769.4 |
| SMAD4 MOUSE | GAGAACATTGGATGGACGAC ACATACTTGGAGCATTACTCTG | 242 | 54 | 40 | NCBI Reference Sequence: NM_001364967.1 |
| SMAD7 MOUSE | ACTCTGTGAACTAGAGTCTCCC CTCTTGGACACAGTAGAGCCT | 241 | 59 | 40 | NCBI Reference Sequence: NM_001042660.1 |

Following that, the created technique was utilized to synthesize complementary-DNA using iScript Reverse Transcription Supermix for RT-PCR (170-8841; Bio-Rad, Hercules, CA, USA). The gene's expression was determined using SsoFast Evagreen Supermix and qRT-PCR (GTC96S, Clever) in triplicate (172-5200; Bio-Rad, Gladesville, NSW, Australia) (*Lister et al., 2020*; *Hidayat et al., 2022*). Table 1 shows the primer sequence (Macrogen) and Table 2 shows the RNA concentration/purity.

## Mesangial cell's morphology observation

The mesangial cells morphology was observed using an inverted microscope (CKX41-F32FL, Olympus). The observed parameter included cells proliferation that was indicated by the density of the cells. The morphology of the treated mesangial cell group was compared to the non-treated mesangial cell as the control group.

## Statistical analysis

To verify the results of different treatments, one-way Analysis of Variance (ANOVA) was employed, followed by Tukey HSD, and Dunnet-T3 post-hoc test to confirm significant differences for all treatments ($P < 0.05$). The non-parametric test was evaluated by Mann-Whitney test ($p < 0.05$). The data were analyzed using the Statistical Package for the Social Sciences (SPSS) software version 16.

## RESULTS

### TGF-β1 level

The level of TGF-β1 of quercetin treatments is shown in Fig. 1. The lowest TGF-β 1 (79.77 pg/mL) was the CDK cells model treated with 5 μg/mL of quercetin. Quercetin reduced the TGF-β1 level significantly compared to the CDK cells model according to the statistical analysis ($p < 0.05$).

### TNF-α level

The result data show that quercetin decreased TNF-α level significantly from the positive control (Fig. 2). The lowest TNF-α level (154.97 pg/mL) was obtained when the CDK cells

Table 2 **RNA concentration and purity.**

| Sample | Concentration of RNA (ng/μL) | RNA Purity (260/280 nm) |
|---|---|---|
| Negative Control | 43.37 | 2.0776 |
| Positive Control | 60.16 | 2.2706 |
| Quercetin 50 μg/mL | 18.65 | 1.9768 |
| Quercetin 10 μg/mL | 53.87 | 1.7291 |
| Quercetin 5 μg/mL | 52.07 | 1.7816 |

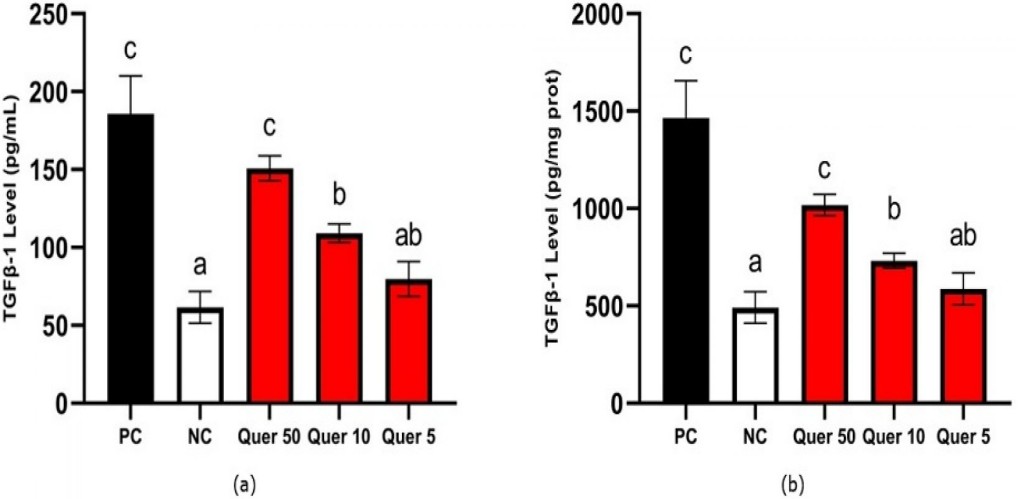

Figure 1 **TGF-β1 level of quercetin treatment on CKD cell model.** (A) TGF-β1 level (pg/mL) on CKD cell model, (B) TGF-β1 level (pg/mg protein) on CKD cell model. *The data is shown in from a mean value with a standard deviation, $n = 3$. PC: Positive control/glucose-induced mesangial cells; NC: Negative control/untreated mesangial cells, Quer 50: Positive control + Quercetin 50 μg/mL, Quer 10: Positive control + Quercetin 10 μg/mL, Quer 5: Positive control + Quercetin 5 μg/mL Different letter (a,ab,b,c) show significant difference among treatment on TGF-β1 level pg/mL (Fig. 1A), different letter (a,ab,b,c) indicate significant difference among treatment on TGF-β1 level pg/mg protein (Fig. 1B) based on Tukey's HSD post hoc test ($p < 0.05$).

model was treated with 5 μg/mL of quercetin. Quercetin reduced TNF-α levels significantly compared to CKD cells model ($p < 0.05$).

## MDA level

Figure 3 shows MDA level of quercetin treatment. Quercetin could reduce MDA levels to 758.72, 690.76, and 579.70 ng/mL at 50, 10, and 5 μg/mL concentrations, respectively. Quercetin reduced MDA level significantly compared to the CKD cells model according to the statistical analysis ($p < 0.05$).

## Gene expression of SMAD7, SMAD3, SMAD2, SMAD4

Figure 4 show the results of gene expression values. The results indicated that quercetin for all concentrations tests (50, 10, and 5 μg/mL, respectively) significantly decreased ($p < 0.05$) the relative ratio of SMAD2 gene expression to 13.06, 7.84, and 5.42 compared

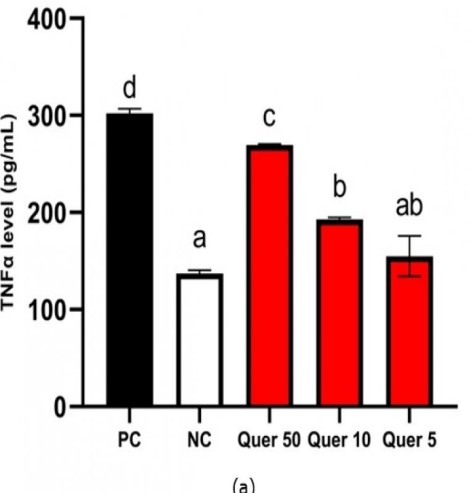
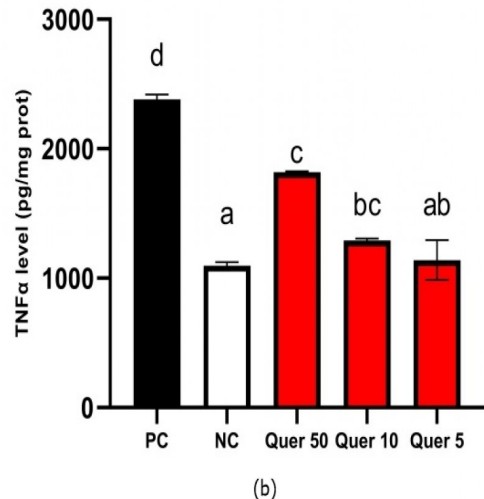

**Figure 2** **TNF-α level of quercetin treatment on CKD cell model.** (A) TNF-α level (pg/mL) on CKD cell model. (B) TNF-α level (pg/mg protein) on CKD cell model. *The data is shown in from a mean value with a standard deviation, $n = 3$. PC: Positive control/glucose-induced mesangial cells; NC: Negative control/untreated mesangial cells, Quer 50: Positive control + Quercetin 50 μg/mL, Quer 10: Positive control + Quercetin 10 μg/mL, Quer 5: Positive control + Quercetin 5 μg/mL. Different letter (a,ab,b,c,d) show significant difference among treatment on TNF-α level pg/mL (Fig. 2A), different letter (a,ab,bc,c,d) indicate significant difference among treatment on TNF-α level pg/mg protein (Fig. 2B) based on Dunnett T3 post hoc test ($p < 0.05$).

to positive control at 27.76. The treatment decreased the relative ratio of SMAD3 gene expression to 22.93, 15.98, and 7.91 compared to positive control at 45.32. Moreover, quercetin decreased the relative ratio of SMAD4 gene expression to 1.82, 1.66, and 1.15 compared to positive control at 2.23. Meanwhile, quercetin at 10 mg/mL and 5 mg/mL concentrations tests increased the relative ratio of SMAD7 gene expression to 0.71, and 0.92 from the positive control at 0.49.

## Mesangial cell's morphology

The morphology of mesangial cells in this research is shown in Fig. 5. The glucose-induced mesangial cells showed differences between positive control and negative control (normal cell). The research showed that mesangial cells induced by glucose (positive control) caused visible high proliferating cells compared to mesangial cells without glucose induction (negative control). The treatment with quercetin showed that CKD cells model treated with quercetin was close to negative control cell's condition which was characterized by normal cell growth.

## DISCUSSION

CKD is a condition when the function of the kidney is reduced. Quercetin is a natural compound having a wide range of biological effects. Quercetin has been shown to have antioxidant, anticancer, anti-aggregation, anti-inflammatory, and vasodilation activities.

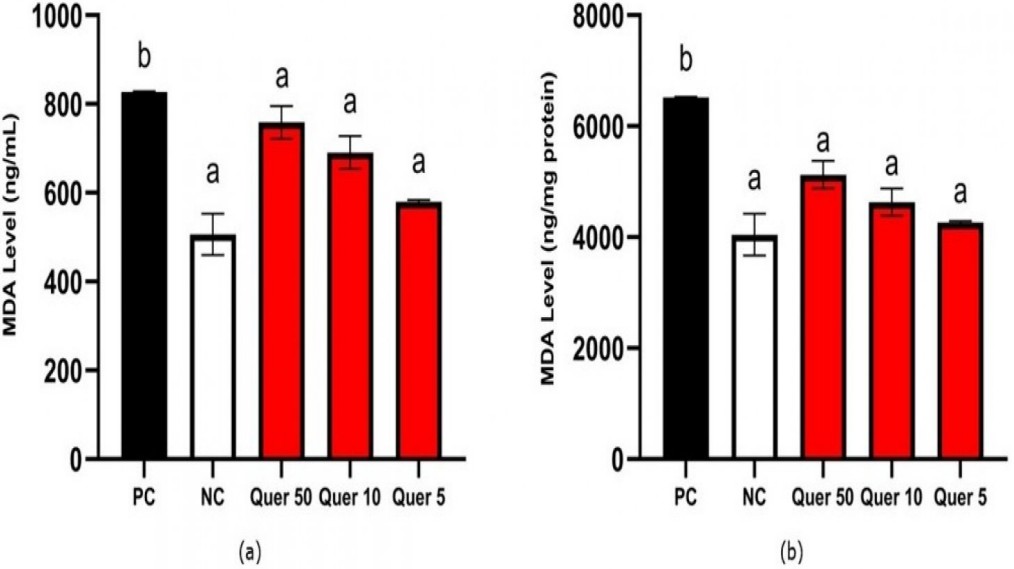

**Figure 3** **MDA level of quercetin treatment on CKD cell model.** (A) MDA level (ng/mL) on CKD cell model. (B) MDA level (ng/mg protein) on CKD cell model. *The data is shown in from a mean value with a standard deviation, $n = 3$. PC: Positive control/glucose-induced mesangial cells; NC: Negative control/untreated mesangial cells, Quer 50: Positive control + Quercetin 50 $\mu$g/mL, Quer 10: Positive control + Quercetin 10 $\mu$g/mL, Quer 5: Positive control + Quercetin 5 $\mu$g/mL Different letter (a,b) indicate significant difference among treatment on MDA level pg/mL (Fig. 3A), pg/mg protein (Fig. 3B) based on Mann Whitney post hoc test ($p < 0.05$).

The goal of this study is to see if quercetin can help in CKD treatment (*Rusmana et al., 2017*; *Lu et al., 2018*, *Yang et al., 2018*, *Perdhana & Suzana, 2019*).

TNF-α is a powerful pro-inflammatory cytokine that aids the immune system during periods of inflammation (*Nair et al., 2006*). Pro-inflammatory cytokine influenced a lot in chronic inflammatory disorders by oxidative stress, and one of the highest is TNF-α. The previous study has shown that quercetin inhibits the production of TNF-α affected by the inhibition of NF-kB immunomodulatory cytokine transcription (*Asni et al., 2009*). Flavonoids can influence the immune response. Quercetin has anti-inflammatory properties by a decrease in the TNF-α production (*Nair et al., 2006*). The result data show that quercetin in all concentrations reduced TNF-α levels in CKD cells model.

MDA is a chemical that can be used to determine how free radicals behave in cells. MDA is usually used as a free radical indicator (*Hojs et al., 2020*; *Ginting et al., 2021*). Another research confirms this argument by claiming that the MDA mediator is a final fat peroxidation product that can define the degree of oxidative stress as a biological biomarker of fat peroxidation (*Tualeka et al., 2019*). The result shows that the level of MDA was decreased by quercetin treatment in CKD cell model. Quercetin has antioxidant activity by regulating glutathione (GSH). Previous researches have found a substantial association between GSH and MDA, and a reciprocal interaction between the two are backwards (*Lin et al., 2017*; *Hojs et al., 2020*). Furthermore, quercetin modulates enzymes

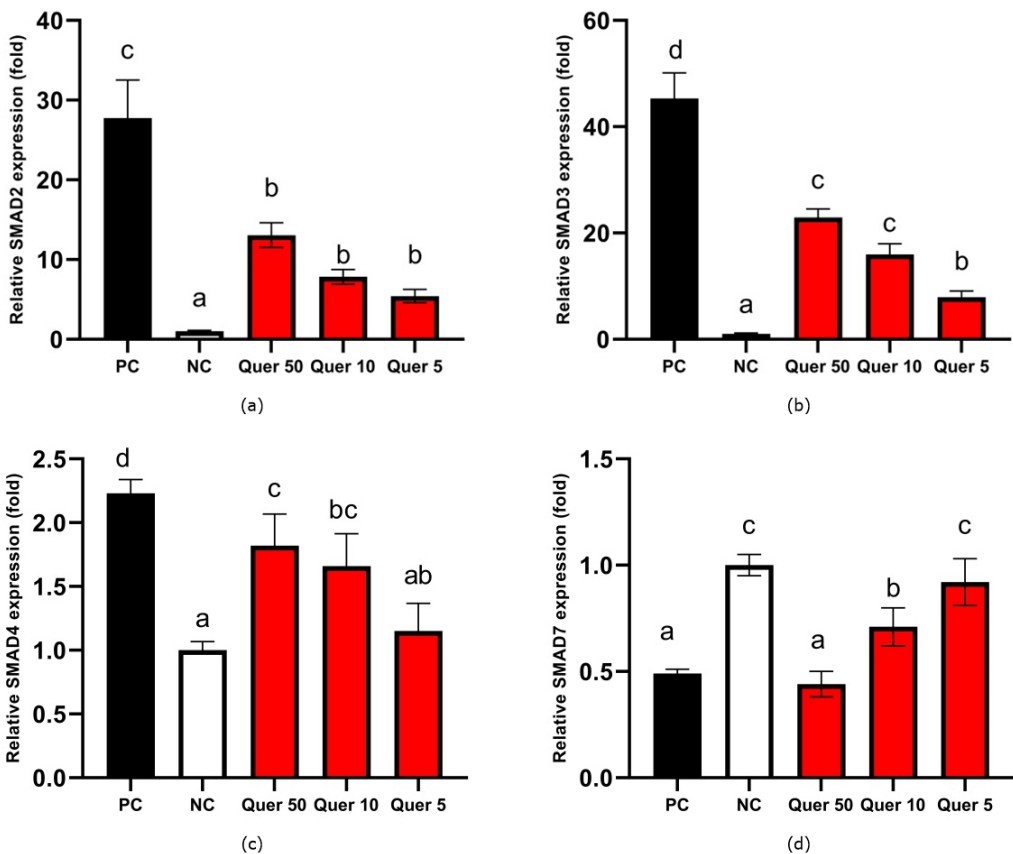

**Figure 4** **SMADs gene expression of quercetin treatment on CKD cell model.** (A) SMAD2 gene expression on glucose-induced mesangial cells. (B) SMAD3 gene expression on glucose-induced mesangial cells. (C) SMAD4 gene expression on glucose-induced mesangial cells. (D) SMAD7 on glucose-induced mesangial cells. An asterisk (*) indicates that the data is shown from a mean value with a standard deviation, $n = 3$. PC: Positive control/glucose-induced mesangial cells; NC: Negative control/untreated mesangial cells, Quer 50: Positive control + Quercetin 50 μg/mL, Quer 10: Positive control + Quercetin 10 μg/mL, Quer 5: Positive control + Quercetin 5 μg/mL Different lowercase letters (a, b, c) indicate significant difference among treatment on SMAD2 gene expression (Fig. 4A), different lowercase letters (a, b, c, d) indicate significant difference among treatment on SMAD3 gene expression (Fig. 4B), different lowercase letters (a, ab, bc, c, d) indicate significant difference among treatment on SMAD4 gene expression (Fig. 4C), different lowercase letters (a, b, c) indicate significant difference among treatment on SMAD7 gene expression (Fig. 4D) based on Tukey HSD post hoc test ($p < 0.05$).

or antioxidant substances to enhance antioxidant properties by affecting signal transduction pathways, thus preventing free radical production (*Lu et al., 2018*).

TGF-β1 is a potent cytokine in playing a driving role in the development of fibrosis by promoting myofibroblasts formation. TGF-β signals mediate the SMADS protein family, namely SMAD2, SMAD3, and SMAD4 (*Meng, Chung & Lan, 2013*). Except for SMAD7 gene expression, glucose stimulation in the positive control boosts all SMADS gene expression. The SMAD2, SMAD3, and SMAD4 genes, notably SMAD2, are upregulated in glucose-induced mesangial cells. Both SMAD2 and SMAD3 are extensively proof-activated in fibrotic kidneys in people and animal models with CKD (*Quezada et al., 2012*). The

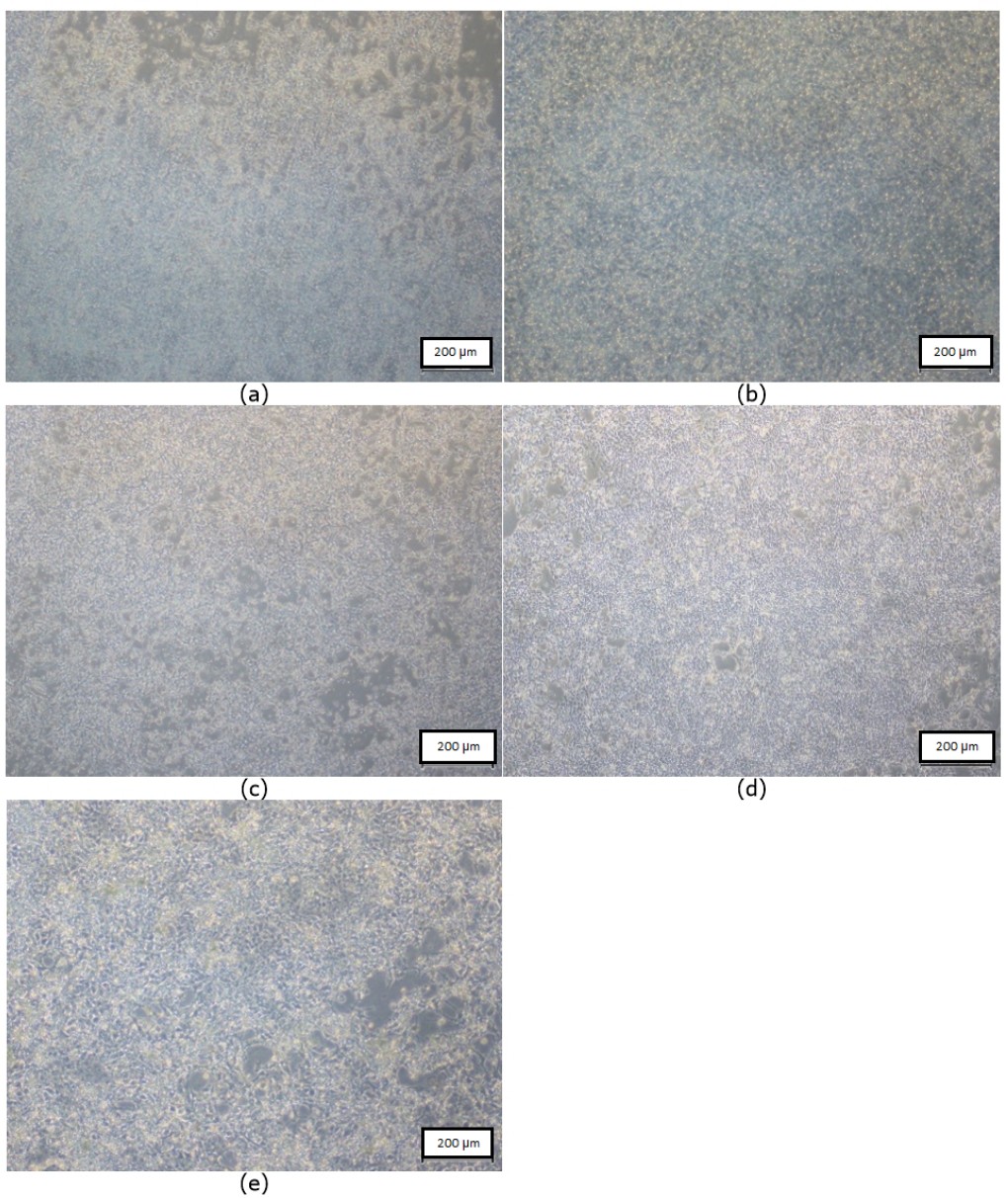

**Figure 5** **Morphology of quercetin treatment on CKD cell model.** (A) Negative control: untreated mesangial cells (B) Positive control: glucose-induced mesangial cells (C) Quer 5: Positive control + quercetin 5 μg/mL (D) Quer 10: Positive control + quercetin 10 μg/mL (E) Quer 50: Positive control + quercetin 50 μg/mL.

expression of SMAD3, SMAD2, and SMAD4 genes increases with increasing TGF-β (*Wang et al., 2018*). This is in line with the findings of this investigation, which is increasing the expression of SMAD2, SMAD3, and SMAD4 genes in accordance with increasing levels of TGF-β. Previous studies have shown that poricoic acid compound can inhibit the phosphorylation of SMAD3 induced by TGF-β induction (*Wang et al., 2018*). Poricoic acid also protects the kidneys during the progression of Acute Kidney Injury (AKI) to

CKD by regulating the Gas6/AxlNFkB/Nrf2 pathway. Poricoic acid upregulates Gas6/Axl signaling in AKI, reducing oxidative stress and inflammation, and reducing renal fibrosis by downregulating Gas6/Axl signaling in CKD (*Chen et al., 2019*). Poricoic acid ZA further suppresses TGF-β1/SMADs pathway through inhibiting SMAD2/3 phosphorylation by blocking SMAD2/3-TGF-β RI protein interaction (*Wang et al., 2017*). This result is in accordance with previous studies that found decreases in TGF-β1, SMAD2, SMAD3, and SMAD4 after the treatment. TGF-β1 is inhibited by SMAD7, which is a nonspecific antagonist. SMAD7 is considered to block TGF signaling by competing with SMAD2/3 for type I receptor binding, increasing SMAD complex ubiquitination, and suppressing SMAD/DNA complexes in the nucleus (*Chen et al., 2018a*; *Chen et al., 2018b*). Thus, decreasing TGF-β1 causes an increase the SMAD7 gene expression (*Hidayat et al., 2022*). Quercetin inhibits fibrosis incidence through TGF-β /SMADs pathway signaling.

As explained before, the anti-inflammatory drug and antioxidant could be used as CKD therapy. Tripterygium Wilfordii Polyglycosides (TWPs) is the main compound of *Tripterygium Wilfordii Hook. F.* (TWHF) showing anti-fibrosis activity by reducing IL-1, IL-17, interferon-γ (IFN-γ), and TNF-α (*Liu et al., 2021*). Previous studies show that *Abelmoschus manihot* reduces proteinuria and lessens kidney damage and tubulointerstitial fibrosis so that it improves kidney function. These beneficial effects are related to the anti-inflammatory and antioxidant properties of *A. manihot* (*Cai et al., 2017*; *Li et al., 2019a*; *Li et al., 2019b*; *Liao et al., 2019*). Compounds that have good anti-inflammatory and antioxidant properties can act as antifibrosis in CKD therapy. According to the present study, quercetin has antioxidant activity by decreasing MDA level and has anti-inflammatory activity by decreasing TNF-α level. Therefore, quercetin can act as anti-fibrosis with its anti-inflammation and antioxidant activities.

The cells morphology shows that negative control had a confluence of 70–80% for the glucose-induced high proliferation of mesangial cells. Quercetin treatment could reduce cells proliferation. It was indicated by the lower density of the cells compared to that of positive control. The higher quercetin concentration showed higher inhibition to cell proliferation. Quercetin 10 μg/mL was more active to inhibit mesangial cells compared to quercetin 5 μg/mL; while the highest inhibition was shown by quercetin 50 μg/mL. That was indicated by the cell's density being higher compared to negative control with cells density of 70–80% confluence. The previous studies showed that TGF-β1 could induce fibroblast proliferation (*Kamejima et al., 2019*; *Prahastuti et al., 2019*). The results of this study proved that quercetin was able to inhibit the cells proliferation of CKD cells model by reducing the TGF-β1 level. The proposed pathway of quercetin which prevents CKD by regulating inflammation and TGF-β1/SMADS pathway can be seen at Fig. 6.

## CONCLUSION

In conclusion, this work has proven that quercetin has anti-inflammation, antioxidant, and antifibrosis properties in the CKD cells model. Thus, quercetin is a promising compound for CKD therapy and further research in CKD animal models is suggested to be examined.

 

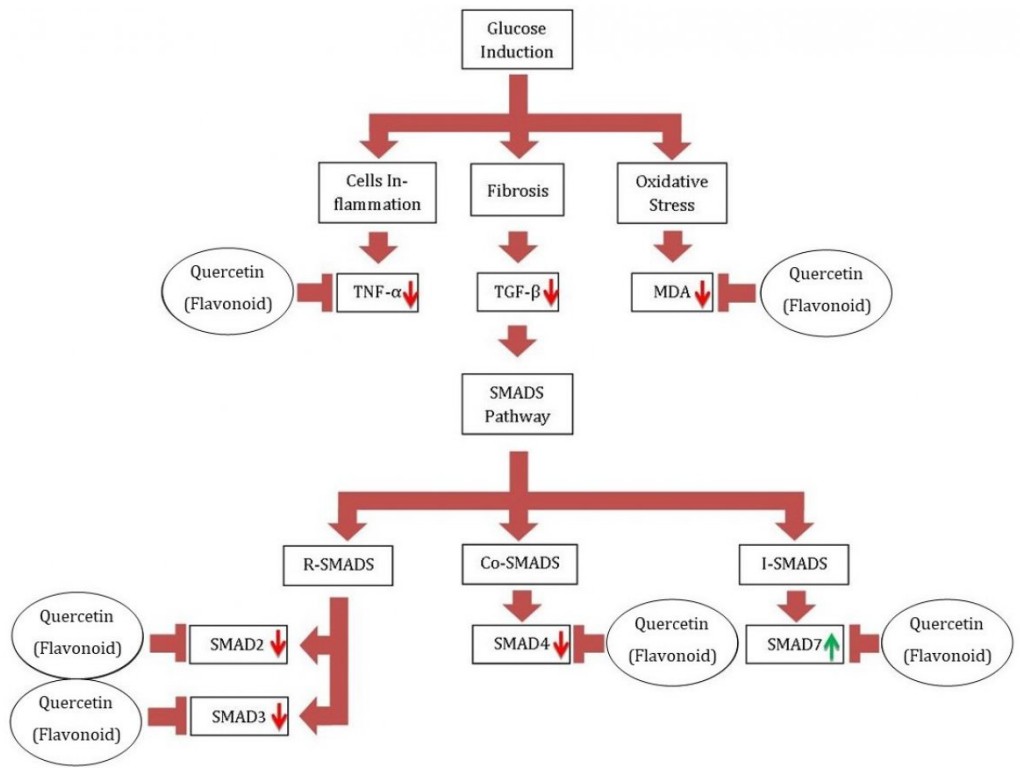

**Figure 6  Proposed pathway of quercetin in preventing CKD.** *Glucose induction could cause the inflammation and oxidative stress on mesangial cells. Cells inflammation induce the level of TNF-α, as oxidative stress increases the level of MDA on the cells, and cells fibrosis increase the level of TGF-β1. The increase of TGF-β1 level induce the expression of R-SMADS and Co-SMADS, while repress the I-SMADS expression. On glucose-induced mesangial cells, quercetin therapy reduced the levels of TGF-B1, TNF-A, and MDA. As it represses TGF-1, quercetin also reduces SMAD2, SMAD3, SMAD4, and increases SMAD7 expression. R-SMADS: receptor-regulated SMADS (SMAD 1, 2, 3, 5, and 8). C0-SMADS: common-mediator SMADS (SMAD4). I-SMADS: Inhibitor SMADS (SMAD 6 and 7).

## SIGNIFICANCE

Previous researches have demonstrated that quercetin suppressed a TNF-α production. Quercetin has been shown to have antioxidant properties by modulating GSH. The antioxidant activity of quercetin was found to lower MDA levels in CKD models in this investigation. Our findings also reveal new information on quercetin's role in preventing fibrosis through TGF/SMAD signaling.

## ACKNOWLEDGEMENTS

We extend appreciation to Meganita Marthania, Cahyaning Riski Wijayanti, Agung Novianto, and Afif Yati for valuable help as research assistants.

## Funding
The authors received no funding for this work.

## Competing Interests
The authors declare there are no competing interests.

## Author Contributions
- Wahyu Widowati conceived and designed the experiments, authored or reviewed drafts of the paper, and approved the final draft.
- Sijani Prahastuti conceived and designed the experiments, authored or reviewed drafts of the paper, and approved the final draft.
- Rita Tjokropranoto conceived and designed the experiments, authored or reviewed drafts of the paper, and approved the final draft.
- Philips Onggowidjaja conceived and designed the experiments, authored or reviewed drafts of the paper, and approved the final draft.
- Hanna Sari Widya Kusuma analyzed the data, prepared figures and/or tables, authored or reviewed drafts of the paper, and approved the final draft.
- Ervi Afifah performed the experiments, analyzed the data, prepared figures and/or tables, authored or reviewed drafts of the paper, and approved the final draft.
- Seila Arumwardana performed the experiments, analyzed the data, prepared figures and/or tables, authored or reviewed drafts of the paper, and approved the final draft.
- Muhamad Aldi Maulana performed the experiments, analyzed the data, prepared figures and/or tables, authored or reviewed drafts of the paper, and approved the final draft.
- Rizal Rizal analyzed the data, authored or reviewed drafts of the paper, and approved the final draft.

## Data Availability
The raw measurements are available in the Supplementary File.

## Supplemental Information
Supplemental information for this article can be found online at http://dx.doi.org/10.7717/peerj.13257#supplemental-information.

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
