# Peer review of "Quercetin prevents chronic kidney disease on mesangial cells model by regulating inflammation, oxidative stress, and TGF-β1/SMADs pathway"

_PeerJ, doi:10.7717/peerj.13257_

## Round 0.1 · original submission · Major Revisions

Interesting experimental research, but some major revisions are required (please, see the reviewers' comments carefully).

Reviewer 1 ·

Basic reporting

The Introduction needs more detail. Specifically, I suggest that the authors need to improve the description regarding the role of Quercitin in CKD to provide more justification for your study, referring to a recent study “Quercetin Treatment Improves Renal Function and Protects the Kidney in a Rat Model of Adenine-Induced Chronic Kidney Disease” by Hu Yang et al. 2018. Moreover, it could be interesting that the authors deepened the theme of CDK; an interesting and recent paper regarding this could be “The Protective Role of Prolyl Oligopeptidase (POP) Inhibition in Kidney Injury Induced by Renal Ischemia–Reperfusion” by Giovanna Casili et al., 2021.
The English language should be improved to ensure that an international audience can clearly understand the text. I suggest to contact a professional editing service to improve English language.

Experimental design

The authors performed a rigorous investigation. The methods were described with sufficient detail and information to replicate.

Validity of the findings

Conclusions are well stated, linked to original research question and limited to supporting results.

Additional comments

Conclusions are well stated, linked to original research question and limited to supporting results.

Reviewer 2 ·

Basic reporting

See the additional comments.

Experimental design

See the additional comments.

Validity of the findings

See the additional comments.

Additional comments

In this work, the authors demonstrated Quercetin prevents chronic kidney disease on mesangial cells model through regulating inflammation and TGF-β1/Smad pathway. Several suggestions are made as follows to improve the quality of the manuscript.
1. Title is not suitable. Please change title to “Quercetin prevents chronic kidney disease on mesangial cells model through regulating inflammation and TGF-β1/Smad pathway”.
2. Natural products or traditional Chinese medicine in Chinese herbal compound prescription and isolated compounds has been applied to treatment of CKD through regulating various molecular mechanisms including inflammation and oxidative stress, TGF-β1/Smad, Wnt/β-catenin pathway reported by many publications such as Trends Pharmacol Sci 2018,39(11):937-952; Front Pharmacol 2021,12:655372; Med Res Rev 2020,40(1):54-78. Please summarize pharmacological effects of natural products or traditional Chinese medicine in the introduction section to improve manuscript.
3. Chemicals and Reagents should be provided in this section. Please include both the manufacturer's name and location (including city, state, and country) for specialized equipment and reagents throughout the manuscript.
4.
5. TCM or natural products against CKD by TGF-β1/Smad pathway have been widely reported by several publications, such as Br J Pharmacol 2018,175(13):2689-2708; J Agric Food Chem 2018,66(8):1828-1842; Phytomedicine 2021,93:153774; Phytomedicine 2018,50:50-60; Free Radic Biol Med 2019,134:484-497. Please compare them with quercetin discuss them in the discussion section.
6. TCM or natural products against fibrosis by regulating inflammation and oxidative stress have been widely reported by several publications, such as; Front Med 2021,8:747922; Frontiers Pharmacol 2021,12:800522; Front Pharmacol 2021,12:774414; Phytomedicine 2020,79:153352; Front Pharmacol 2021,12:630210. Several the latest reviews have summarized the application of natural products in renal injury. Please compare them with quercetin discuss them in the discussion section.
7. Scale bar should be provided in Figure 5.
8. The text must be extensively revised by a native English speaker.

---

## Round 0.2 · accepted · Accept

The authors' responses to the reviewers are satisfactory. No further revisions are needed.

Reviewer 2 ·

Basic reporting

Basic reporting is good.

Experimental design

Experimental design is good.

Validity of the findings

Experimental design is good.

Additional comments

The authors have improved the manuscript, so I suggest the manuscript should be accepted for publication.